# Challenges Regarding the Value of Routine Perioperative Transesophageal Echocardiography in Mitral Valve Surgery

**DOI:** 10.3390/diagnostics14111095

**Published:** 2024-05-24

**Authors:** Luminita Iliuta, Madalina-Elena Rac-Albu, Eugenia Panaitescu, Andreea Gabriella Andronesi, Horatiu Moldovan, Florentina Ligia Furtunescu, Alexandru Scafa-Udriște, Mihai Adrian Dobra, Cristina Mirela Dinescu, Gheorghe Dodu Petrescu, Marius Rac-Albu

**Affiliations:** 1Medical Informatics and Biostatistics Department, University of Medicine and Pharmacy “Carol Davila”, 050474 Bucharest, Romania; luminita.iliuta@umfcd.ro (L.I.); eugenia.panaitescu@umfcd.ro (E.P.); mirela.dinescu@umfcd.ro (C.M.D.); dodu.petrescu@umfcd.ro (G.D.P.); marius.rac-albu@umfcd.ro (M.R.-A.); 2Cardioclass Clinic for Cardiovascular Disease, 031125 Bucharest, Romania; 3Nephrology Department, University of Medicine and Pharmacy “Carol Davila”, 050474 Bucharest, Romania; 4Nephrology Department, Fundeni Clinical Institute, 022328 Bucharest, Romania; 5Department of Cardio-Thoracic Pathology, University of Medicine and Pharmacy “Carol Davila”, 050474 Bucharest, Romania; horatiu.moldovan@umfcd.ro (H.M.); alexandru.scafa@umfcd.ro (A.S.-U.); 6Department of Cardiovascular Surgery, Clinical Emergency Hospital, 014461 Bucharest, Romania; 7Academy of Romanian Scientist (AOSR), 050711 Bucharest, Romania; 8Department of Public Health and Management, University of Medicine and Pharmacy “Carol Davila”, 050474 Bucharest, Romania; florentina.furtunescu@umfcd.ro; 9Department of Cardiology, Clinical Emergency Hospital, 014461 Bucharest, Romania; 10Center of Uronephrology and Renal Transplantation, Fundeni Clinical Institute, University of Medicine and Pharmacy “Carol Davila”, 050474 Bucharest, Romania; mihai.dobra@umfcd.ro

**Keywords:** transesophageal echocardiography, perioperative echocardiography, mitral valve surgery, mitral valve disease, customized software application

## Abstract

Background and Objectives: Transesophageal echocardiography (TEE) is considered an indispensable tool for perioperative evaluation in mitral valve (MV) surgery. TEE is routinely performed by anesthesiologists competent in TEE; however, in certain situations, the expertise of a senior cardiologist specializing in TEE is required, which incurs additional costs. The purpose of this study is to determine the indications for specialized perioperative TEE based on its utility and the correlation between intraoperative TEE diagnoses and surgical findings, compared with routine TEE performed by an anesthesiologist. Materials and Methods: We conducted a three-year prospective study involving 499 patients with MV disease undergoing cardiac surgery. Patients underwent intraoperative and early postoperative TEE and at least one other perioperative echocardiographic evaluation. A computer application was dedicated to calculating the utility of each type of specialized TEE indication depending on the type of MV disease and surgical intervention. Results: The indications for performing specialized perioperative TEE identified in our study can be categorized into three groups: standard, relative, and uncertain. Standard indications for specialized intraoperative TEE included establishing the mechanism and severity of MR (mitral regurgitation), guiding MV valvuloplasty, diagnosing associated valvular lesions post MVR (mitral valve replacement), routine evaluations in triple-valve replacements, and identifying the causes of acute, intraoperative, life-threatening hemodynamic dysfunction. Early postoperative specialized TEE in the intensive care unit (ICU) is indicated for the suspicion of pericardial or pleural effusions, establishing the etiology of acute hemodynamic dysfunction, and assessing the severity of residual MR post valvuloplasty. Conclusions: Perioperative TEE in MV surgery can generally be performed by a trained anesthesiologist for standard measurements and evaluations. In certain cases, however, a specialized TEE examination by a trained senior cardiologist is necessary, as it is indirectly associated with a decrease in postoperative complications and early postoperative mortality rates, as well as an improvement in immediate and long-term prognoses. Also, for standard indications, the correlation between surgical and TEE diagnoses was superior when specialized TEE was used.

## 1. Introduction

Perioperative transesophageal echocardiography (TEE) has become an indispensable diagnostic tool in mitral valve (MV) surgery, as underscored by recent guidelines. TEE offers comprehensive insights into MV anatomy, lesion severity, and ventricular performance, aiding surgeons in selecting the most appropriate surgical approach and assessing preoperative risks. Additionally, it is used intraoperatively to monitor the progress of surgery, allowing for adjustments to the initial plan as needed. TEE is also invaluable for evaluating surgical outcomes and for the prompt diagnosis of potential postoperative complications during the immediate postoperative period.

Recent guidelines have highlighted the necessity of intraoperative TEE in a wide range of cardiac surgeries, from routine coronary revascularization to complex valve repairs, combined procedures, and organ transplantation. It is now considered essential for all open heart and thoracic aortic surgical procedures in adults, barring contraindications. TEE is also recommended during coronary artery bypass grafting (CABG) to confirm and refine preoperative diagnoses, detect new or unexpected pathologies, adjust anesthetic and surgical plans, and assess surgical outcomes. [1]

The use of perioperative TEE is increasingly advocated across all practice guidelines to enhance the outcomes of cardiac surgeries. Physicians performing intraoperative echocardiography must fully understand the technical steps of the surgical procedure and potential complications to guide their evaluations and provide pertinent echocardiographic information.

Historically, perioperative echocardiography was primarily conducted by senior cardiologists specializing in this area. However, in recent years, cardiac anesthesiologists have gained considerable proficiency, transforming the landscape of intraoperative management. Despite this shift, the specialized expertise of cardiologists remains crucial, especially considering that perioperative TEE influences surgical plans in 10% to 25% of cases. Effective intraoperative management thus depends on seamless collaboration and communication among cardiologists, surgeons, and anesthesiologists, as well as with the entire surgical team.

The American Society of Anesthesiologists Task Force Guidelines now recommend that anesthesiologists use TEE in all mitral valve surgical procedures to confirm preoperative diagnoses, detect new pathologies, adjust anesthetic and surgical management, and evaluate surgical outcomes.

While most patients undergoing scheduled cardiac surgery undergo specialized preoperative examinations by a senior cardiologist to define the cardiac pathology, intraoperative TEE conducted by the anesthesiologist can refine the surgical strategy and identify any additional or incidental findings that might significantly alter the surgical approach.

The primary goal of the intraoperative examination prior to surgery is to confirm known conditions and exclude any additional pathologies that could affect the surgical plan. For instance, discovering incidental findings like a persistent left superior vena cava (PLSVC) may necessitate adjustments in administering retrograde cardioplegia. Similarly, detecting a patent foramen ovale (PFO) could lead to a reassessment of venous cannulation strategies, particularly if PFO repair is deemed necessary. Additionally, a thorough assessment might uncover severe aortic atherosclerotic disease, potentially influencing decisions on aortic cannulation, cross-clamping, or the use of an intra-aortic balloon pump, depending on the location and severity of the disease.

These routine assessments can be performed by either an anesthesiologist trained in intraoperative TEE or a senior cardiologist specialized in perioperative echocardiography, based on the level of diagnostic detail required and the necessity for guidance during the procedure. The ongoing debate centers on whether specialized examinations should be routinely performed and under what circumstances they are necessary.

Since its initial application in MV surgery in 1979, the role of TEE has become fundamental, establishing itself as an indispensable tool in the perioperative evaluation of MV surgery, including minimally invasive procedures. It is now considered a class I indication for the surgical reconstruction of the mitral valve. TEE is crucial for evaluating MV pathology, grading mitral regurgitation (MR), identifying potential risk factors, and assessing post-repair outcomes. Real-time three-dimensional TEE provides essential anatomical visualization of the MV apparatus, which is vital for virtual surgical planning and determining the appropriate annuloplasty ring size [2,3,4].

The execution of intraoperative TEE is particularly challenging due to the poor hemodynamic status of patients undergoing surgery. Various intraoperative parameters significantly influence the TEE assessment of numerous cardiac lesions, especially in determining the degree of valvular regurgitation [5]. These parameters include preload variability (affected by intravascular volume status, vasodilator treatment, and general anesthesia factors), afterload variability (influenced by vasopressor and inotropic treatments, vasodilator therapy, the presence of an intra-aortic balloon pump, LVOT (left ventricular outflow tract) obstruction, and other anesthesia-related factors), as well as the presence of arrhythmias and myocardial function, which can vary post hypothermia or following cardioplegic solution administration [6].

In the operating room, the echocardiographer has the critical task of interpreting various parameters and integrating them with the specifics of each case to provide the surgeon with more accurate data [7].

Postoperatively, the condition of patients undergoing cardiac surgery differs significantly from their preoperative state, necessitating that echocardiography be conducted by a trained specialist. During the ICU stay, the importance of diagnosing diastolic dysfunction becomes more pronounced. Despite our review of the current literature for standardized diagnostic criteria for this patient group in the ICU, the diagnosis and treatment of this condition remain challenging and often unsatisfactory [8,9]. The clinical similarities between diastolic and systolic heart failure underscore the necessity for specialized postoperative TEE to differentiate these conditions and to develop standardized management algorithms [10,11].

Recent literature and guidelines have not conclusively determined whether perioperative TEE should be routinely performed by a senior cardiologist trained for both the operating room and ICU environments, or if it can be effectively managed by an anesthesiologist with specialized examinations reserved for special conditions.

Furthermore, existing studies have not adequately assessed the utility of routinely specialized perioperative TEE in MV surgery, nor the correlation between surgical and preoperative TEE diagnoses.

### Aim of the Study

The primary objective of this study is to evaluate the utility of perioperative TEE when performed by a senior cardiologist specialized in this imaging technique in MV surgery. The secondary objective is to determine the correlation between surgical outcomes and the diagnoses obtained by specialized or routine intraoperative TEE.

Ultimately, this study aims to establish practical guidelines for when specialized perioperative TEE, performed by a senior cardiologist, is most useful compared to its routine use by an anesthesiologist in MV surgery.

## 2. Materials and Methods

### 2.1. Study Population, Setting, and Data Collection

We conducted a prospective study over three consecutive years at the “C.C. Iliescu” Institute for Cardiovascular Diseases, Bucharest, involving 499 patients with mitral valve (MV) disease undergoing cardiac surgery. Patients were divided into two groups based on the type of perioperative transesophageal echocardiography (TEE) performed:-Group A—125 patients who underwent specialized TEE either intraoperatively before or after cardiopulmonary bypass (CPB) or early postoperatively.-Group B—374 patients who did not undergo specialized TEE, with perioperative evaluations performed by a trained anesthesiologist when necessary.

The study included demographic data, details on valvular lesions, specific indications for performing perioperative TEE, its impact on surgical decision making during and immediately post surgery, and the immediate postoperative outcomes. All patients in Group A also underwent routine TEE evaluations performed by an anesthesiologist.

### 2.2. Ultrasound Methods

Patients were evaluated by TEE intraoperatively and early postoperatively, with at least one echocardiographic evaluation performed perioperatively. All examinations were conducted using a portable General Electric VIVID i ultrasound machine. Our techniques and calculations adhered to the guidelines of the European and American Societies of Echocardiography [11,12].

Specialized TEE focused on evaluating key parameters such as valve morphology and function, heart cavity dimensions (including end-systolic and end-diastolic volumes of the left ventricle, left atrium diameter, and indexed volumes), and both systolic and diastolic performances of the left ventricle using Tissue Doppler Imaging techniques. Pulmonary artery pressure was also estimated [13,14,15].

Data were collected using a dedicated software application specifically designed for this study. The primary preoperative variables input into the application included patient ID, age, gender, preoperative diagnosis, surgical risk (using EUROSCORE II for risk stratification), and type of surgical intervention.

The surgical procedures were categorized based on the type of valvular lesion and the corresponding surgical intervention. Figure 1 depicts the structural organization of the study group. Additionally, we analyzed how intraoperative adjustments made based on specialized TEE findings influenced the surgical strategy and how postoperative evaluations with specialized TEE in the ICU impacted subsequent treatment decisions. We also assessed variables related to surgical performance (such as surgery duration and intraoperative complications) and postoperative outcomes (including complication types, mortality occurrences, and their underlying causes).

### 2.3. Statistical Analysis

Variables were summarized as frequencies and percentages. We performed statistical correlations between the diagnoses from specialized and routine TEE using the McNemar test, with a significance threshold set at *p* < 0.05 for all analyses. Statistical analyses were conducted using SPSS version 23.0.

Additionally, we assessed the utility and benefits of perioperative specialized TEE when used in conjunction with routine TEE in MV surgery. The software application used for this analysis incorporated medical data from our database as well as economic data related to the costs of specialized TEE, which were provided by the specialized departments of our institute. 

Mortality rates following surgical interventions were analyzed by comparing the outcomes in patients undergoing routine versus specialized TEE. We calculated the mortality rates for each subgroup, taking into account the correlation coefficients between the diagnoses from the two types of TEE.

The application evaluated the utility of specialized TEE by considering parameters related to the procedure, the surgical intervention, and patient characteristics. These parameters included the number of specialized TEEs per patient, whether the information from specialized perioperative TEE altered the therapeutic algorithm or surgical intervention, and the correlation between diagnoses from routine and specialized TEE with the surgical diagnosis. The application quantified the utility of specialized perioperative TEE for each patient group using a scoring algorithm based on the following criteria (1 point awarded for each positive response):-Did information from TEE alter the therapeutic algorithm or surgical intervention?-Was the correlation coefficient between routine and specialized TEE strong (Pearson correlation coefficient greater than 0.5 earns 0 points), medium (Pearson correlation coefficient between 0.39 and 0.5 earns 1 point), or poor (Pearson correlation coefficient less than 0.3 earns 2 points)?-How was the correlation between the diagnosis from specialized or routine TEE and the surgical diagnosis rated: very good (3 points), good (2 points), medium (1 point), or poor (0 points)?-Was the early postoperative specific mortality rate lower in patients who underwent specialized intraoperative TEE?-Were early postoperative complication rates higher in patients who received specialized postoperative TEE in the ICU? (Also analyzed by type of surgical intervention and risk groups).

These criteria were designed to systematically evaluate the impact of specialized TEE on patient outcomes and surgical efficacy.

To calculate the specific mortality rates for each subgroup of patients, including TEE indications and diagnoses, the software application considered various parameters, such as age, gender, co-morbidities, and associated risk factors.

The application computed a benefit score by summing the values that corresponded to the answers to the specified questions. This score was then used to estimate the utility of specialized perioperative echocardiography across different types of indications and surgical interventions. Data were categorized based on the type of surgical lesion and the interventions, according to the level of exposure to risk factors. For each level of exposure, the number of patients who underwent specialized perioperative echocardiography (cases) and those who did not (controls) were recorded. Confounders were addressed through stratification. Data interpretation was guided by the following hypotheses:-A lower benefit score for specialized TEE was deemed economically unfavorable, suggesting an uncertain indication for specialized perioperative echocardiography in these patients.-A benefit score for specialized TEE that was equal to that of routine TEE placed patients in a subgroup classified with a relative indication for performing specialized perioperative echocardiography, with risks and benefits evaluated on a case-by-case basis.-A higher benefit score for specialized TEE was considered economically favorable. In these cases, specialized perioperative echocardiography was recommended as a standard indication for all relevant patients.

## 3. Results

The 499 patients who were the subject of the study were classified according to the surgical intervention as follows:-Five patients underwent mitral valvuloplasty (1%);-A total of 392 patients (78.56%) underwent mitral valve replacement (MVR) with metallic prosthesis (345 patients with bi-leaflet prosthesis and 47 patients with mono-leaflet prosthesis);-A total of 102 patients (20.44%) underwent MVR associated with another valve replacement (mitral and aortic valve replacements in 78 patients—15.63%; mitral and tricuspid valve replacements in 12 patients—2.4%; and mitral, aortic, and tricuspid valve replacements in 12 patients—2.4%) (Figure 1).

To establish the indications for specialized TEE depending on the type of surgical intervention, we considered the intraoperative and early postoperative period separately. The main indications for specialized intraoperative TEE were (Figure 2):-To correct and complete the preoperative diagnosis regarding the mechanism and severity of the mitral valve lesion, and thus establish the opportunity for mitral valvuloplasty or MVR;-To evaluate residual MR and the need for reintervention of iterative mitral valvuloplasty or MVR in mitral valvuloplasty;-To confirm or deny the presence of associated valvular lesions;-For routine control of prosthesis function;-To evaluate acute, life-threatening hemodynamic dysfunction after CPB.

Detailed analysis in patients with surgical MV lesions who underwent specialized perioperative TEE showed a low efficiency of this investigation in patients who underwent isolated MVR. The benefit associated with the use of this specialized diagnostic method was superior for the patients who underwent mitral valvuloplasty and associated valve replacement (MVR with aortic or tricuspid valve replacement or triple valve replacement).

All mitral valvuloplasty patients underwent specialized intraoperative TEE both before CPB and after surgery to evaluate residual mitral regurgitation (MR) and to assess the need for re-intervention of iterative mitral valve repair or MVR. The benefit score for this investigation was higher for specialized TEE, but without statistical significance due to the small number of patients. 

Among the 392 patients who underwent isolated MVR, specialized TEE was performed in 126 patients (32.14%), as follows: in 49 patients (12.5%), TEE was requested only before CPB; in 30 patients (7.65%), TEE was performed both before and after CPB; and in 47 patients (11.99%), TEE was performed only after CPB.

Specialized TEE was performed only before CPB in 27 patients (6.89%), and it was indicated for a correct and complete preoperative diagnosis of the mitral lesion mechanism and severity regarding the need for mitral valvuloplasty (8 patients) or MVR. The indication for specialized TEE in this situation had an overall lower benefit ratio for specialized TEE; it was unfavorable for these patients and it did not provide supplementary significant information compared with preoperative TTE. However, in the subgroup of patients with ischemic MR also undergoing CABG, this investigation proved to be very useful with a higher benefit ratio. At the same time, in patients with MR and severe LV systolic dysfunction, specialized intraoperative TEE had a relative indication, with the benefit ratio being equal to that of non-specialized TEE.

The indication of specialized TEE both before CPB for a more accurate diagnosis and after CPB for routine control of prosthesis function (in 30 patients—7.65%) was associated with a lower benefit ratio; routine TEE in this situation was sufficient as an imaging investigation. Also, specialized intraoperative TEE used for suspicion of prosthesis dysfunction (in 4 patients—1.02%) had low utility; the diagnosis made by the anesthesiologist through routine TEE was the same as that made by the senior cardiologist. Specialized TEE involved an additional cost because it overlapped routine TEE and prolonged the duration of the surgical intervention without involving any change in the surgical strategy.

On the other hand, performing specialized TEE intraoperatively, after surgical correction to evaluate acute, life-threatening hemodynamic dysfunctions (in 22 patients—5.61%) had a firm indication, and was associated with a significantly higher benefit ratio. In 20 patients, specialized TEE brought new information revealing the cause of hemodynamic dysfunction and helped in choosing the optimal treatment strategy, with a higher benefit ratio (12 patients with a filling deficit; 7 patients with global LV systolic dysfunction, with aortic counterpulsation being necessary in 5 patients; and 1 patient with RV systolic dysfunction). Only in 2 patients could specialized intraoperative TEE not establish the cause of hemodynamic dysfunction.

Also, the indication of specialized intraoperative TEE after MVR to evaluate associated valvular lesions that may have been underestimated preoperatively due to the MV lesion (21 patients—5.35%) was associated with a higher benefit ratio. Thus, in 16 patients, specialized intraoperative TEE confirmed the presence of a hemodynamically significant associated aortic insufficiency and, in 3 patients, it revealed the presence of a hemodynamically significant associated aortic insufficiency (which was surgically corrected at the same time operatively in 2 patients). In 2 patients, intraoperative TEE confirmed the presence of a tricuspid lesion. In these situations, the benefit ratio associated with specialized intraoperative TEE was higher and this imaging method could be included among the standard indications for these patients.

In 27 patients (6.89%), specialized TEE was requested intraoperatively only before CPB to confirm or deny the presence of associated valvular lesions. In this subgroup, in 7 patients, TEE indicated aortic or tricuspid valvuloplasty or replacement for preoperatively undiagnosed lesions. The information provided by specialized TEE radically changed the surgical strategy (4 patients with associated aortic valve replacement, 3 patients with associated tricuspid valvuloplasty or valve replacement) and the benefit ratio was higher, with specialized TEE having a firm indication.

The main indications for intraoperative specialized TEE provided by a senior cardiologist and the implications of the diagnosis for the surgical strategy and therapeutic approach are summarized in Table 1.

Overall, concordance between the diagnosis provided by specialized TEE prior to CPB and the surgical diagnosis was generally good (r = 0.72, *p* < 0.005) and better than the diagnosis provided by routine TEE in this subgroup of patients (r = 0.49, *p* < 0.05). 

The multivariate regression analysis revealed that most of the discrepancies regarding specialized TEE were related to the description of valve morphology, which impacted the surgical strategy in patients undergoing MV valvuloplasty (r = 0.32, *p* = NS); but, because of the small number of patients with this type of surgical intervention and considering the learning curve, the results were not significant.

Regarding the benefit of early postoperative specialized TEE in the ICU, the benefit ratio did not show good efficiency in terms of performing this investigation by a senior cardiologist, with the experience of the anesthesiologist being sufficient to make a correct diagnosis and less expensive. However, early postoperative specialized TEE in the ICU remained a necessary investigation in certain special situations in which a more detailed examination by an experienced imaging physician was necessary.

In our study group (499 patients), early postoperative specialized TEE was performed in 205 patients (41.08%) among the patients with operated MV lesions. The indications for performing this investigation were not different from the other categories of patients (coronary or other cardiac pathologies patients).

The main indications of early postoperative specialized TEE in the ICU were:-To establish the etiology of acute hemodynamic dysfunction or of hemodynamic instability;-The suspicion of pericardial or pleural effusions or pulmonary embolism;-The suspicion of early mitral prosthesis or aortic prosthesis dysfunction;-To assess the severity of residual MR after mitral valvuloplasty;-To assess the severity of the associated valvular lesion known preoperatively;-To assess RV systolic performance.

The indications for the use of specialized early postoperative TEE in the ICU according to the specific conditions are shown in Figure 3. 

In patients diagnosed with acute spontaneous hemodynamic dysfunction or with hemodynamic instability (95 patients—19.04%), performing specialized TEE was associated with a higher benefit ratio. In 47 of these patients, specialized TEE (with complete tissue Doppler evaluation) showed a filling deficit with severe LV diastolic dysfunction. In another 7 patients with global LV systolic dysfunction, the need for inotropic support was determined, and for another 2 patients with segmental contractility disorders, coronary angiography was necessary. Significant pulmonary hypertension was observed in 5 patients, and pulmonary thromboembolism was observed in 2 patients. For the last 30 patients in this subgroup, pericardial or pleural effusion was observed, which was also evacuated.

Another group of 23 patients (4.61%) who underwent specialized TEE had a suspicion of pericardial or pleural effusion, a suspicion that was confirmed in 18 of them, thus obtaining a higher benefit ratio.

For paraprosthetic leakage due to early disinsertion of prosthetic annulus, performing early postoperative specialized TEE was very useful, being a standard indication. In 18 patients with suspected paravalvular leak, it was confirmed in 2 patients by early postoperative TEE, requiring reintervention, and in 4 patients with early prosthesis disinsertion suspicion, one was confirmed to need reintervention. For 3 patients (0.6%), early postoperative specialized TEE was performed to evaluate the severity of a residual MR after mitral valvuloplasty with a higher benefit ratio.

On the other hand, in patients with pulmonary embolism suspicion (12 patients—2.40%), specialized TEE in the ICU confirmed only one case with an associated lower benefit ratio, and it was considered that this investigation would be less relevant than computed tomography. Also, for 43 patients (8.62%), early specialized postoperative specialized TEE was performed for the suspicion of early mitral prosthesis dysfunction (37) or aortic prosthesis dysfunction (6 of them having two or three valves replaced). Thus, in 19 patients, specialized TEE was requested for early incomplete prosthesis thrombosis, and this hypothesis was confirmed in all patients with a lower benefit ratio. The same pattern was observed for patients with suspected early thrombosis of aortic prosthesis (2 patients). 

Also, performing early postoperative specialized TEE after isolated or associated MVR to assess the severity of the associated valvular lesion known preoperatively (26 patients—5.21%) was associated with a lower benefit ratio. However, specialized early postoperative specialized TEE performed to assess the RV systolic performance (3 patients—0.6%) had a relative indication with a equal benefit ratio.

To conclude, the information presented above regarding the reasons for performing specialized ETE in the early postoperative period, as well as the correlation between the diagnoses made by the anesthesiologist and the senior cardiologist, calculated using the Pearson correlation coefficient, the usefulness of each investigation, and the changes in therapy generated by performing specialized TEE are presented Table 2.

To summarize, the main indications for performing specialized perioperative TEE in patients with MV lesions undergoing cardiac surgery (adapted to the specific conditions in our country) revealed by the benefit analysis were classified as standard, relative, and uncertain indications. Each of these could be divided into intraoperative and immediate postoperative indications and are presented in Table 3. 

## 4. Discussion

Intraoperative TEE serves as a flexible diagnostic and monitoring instrument for guiding patients through various cardiac surgical procedures, particularly in today’s era in which procedures are becoming more complex and less invasive. TEE has become integral in cardiac surgery, driving advancements and interventions. Particularly in minimally invasive cardiac surgery (MICS), TEE plays a pivotal role in mitigating cardiovascular complications. Cutting-edge imaging techniques are vital for the continual expansion of MICS, especially as innovative transcatheter systems emerge. High-quality imaging throughout the perioperative period is crucial for adjusting surgical strategies and enhancing patient results. Additionally, TEE offers the advantage of delivering live updates on intrathoracic structures, aiding intraoperative decision making [15].

Although transcatheter technology is rapidly growing and represents a promising strategy, the surgical approach remains the best way to repair a degenerative mitral valve regurgitation. In this context, robotic surgery is technologically the most advanced method of minimally invasive mitral valve repair [16].

TEE performed during transcatheter structural cardiac interventions may result in greater complications than that performed in the nonoperative setting or even that performed during cardiac surgery. However, there are limited data on complications associated with TEE during these procedures [17].

Even when patients arrive at the operating room with thorough pre-procedural evaluations, conducting a comprehensive echocardiographic examination beforehand in a systematic manner remains crucial [1].

Employing a standardized image acquisition protocol enables an anesthesiologist trained in perioperative TEE to routinely perform this task. They possess the capability to grasp the implications of new findings on the surgical plan and effectively relay echocardiographic results to the surgical team, offering relevant information. It is paramount that the intraoperative TEE anesthesiologist operates as an integrated and engaged member of the cardiac team. In situations where the examination presents challenges, this individual should facilitate a cognitive exchange between specialties and request specialized TEE assessment [4,5]. 

Surgical interventions for mitral valvuloplasty or for MVR need intraoperative TEE both before and after CPB for many reasons. The utilization dynamics vary depending on the type of surgical procedure, with our study findings aligning closely with those reported in the existing literature. In the case of our study, unfortunately, there were few cases (5 patients, 1% of cases) because mitral pathology in our country is mostly rheumatic and is not suitable for valvuloplasty. Thus, specialized TEE was performed in all patients who underwent mitral valvuloplasty both before and after CBP. In the literature, this imaging technique is primarily utilized for monitoring mitral valvuloplasty procedures compared to other surgical interventions. Its usage has shown a positive trend in recent years [8,18]. 

The frequency of requesting specialized TEE during surgical interventions for isolated MVR rose over the course of our study. This increase can be attributed to two significant factors: greater accessibility to this intervention and heightened caution among surgeons and anesthesiologists, who now view this examination as routine for monitoring prosthetic control following the cardiac phase of the intervention. Based on the subgroup analysis in this study, it was noted that TEE was more frequently employed for ischemic MR. Moreover, specialized intraoperative TEE offers vital information that necessitates communication with the surgeon. This includes evaluating heart valve function, identifying abnormalities such as clefts or excessive motion in scallops/segments accompanying sub-valvular structures during diastole, and assessing leaflet appearance, including thickness, masses/vegetations, and calcification [19,20,21]. Additionally, it aids in monitoring cardiac output to guide fluid management, detecting heart wall motion abnormalities, identifying pericardial effusion or cardiac tamponade, and diagnosing aortic dissection. The risk for systolic anterior motion (SAM) after MV repair should be assessed prior to surgical intervention by specialized measurements, including multiplanar reconstruction of 3D datasets. These measurements need a complete and specialized preoperative and intraoperative TEE evaluation.

The severity of functional MR fluctuates with intraoperative hemodynamic changes, particularly under general anesthesia, where it might improve compared to preoperative assessments. Hence, it is crucial to administer intravenous fluids, inotropic agents, and afterload challenges to prevent inaccurate estimations of MR severity. Also, assessment of associated valvular lesions may need specialized intraoperative TEE evaluation [1,4,20]. A comprehensive evaluation of the tricuspid valve, including the tricuspid annulus, should be performed at the time of surgery by a specialized echocardiographist for the evaluation of the severity of functional tricuspid regurgitation and complete evaluation of RV performance.

To address the initial inquiry posed in this article regarding who should perform TEE and under what circumstances, the answer varies depending on the heart conditions outlined in the study. Specialized intraoperative TEE post surgery holds significant importance in evaluating MV repair [20]. A thorough comprehension of the repair method executed by the surgeon, conveyed by the senior cardiologist conducting the TEE, is imperative as the surgical approach influences echocardiographic outcomes. Anesthesiologists can swiftly assess MV repair once the aortic clamp is released, and if any issues are identified, specialized TEE should be promptly conducted. Regarding MVR, our study indicates that routine specialized TEE for assessing prosthetic valves in the mitral position is not warranted due to unfavorable usefulness considerations. Anesthesiologists can adequately assess prosthetic valve motion after the removal of the aortic cross-clamp. Specialized evaluation is only necessary if they identify insufficient function of mechanical disks or bioprosthetic leaflets or the presence of intra- or paravalvular regurgitation. Additionally, in the immediate postoperative period in the ICU, there was variability in specialized TEE indications during the study period, albeit with an overall positive trend. Notably, for triple-valve replacement, a negative trend was observed, attributed to increased intraoperative utilization of TEE and improved surgical outcomes over time. However, the reduced indication for immediate postoperative TEE in the ICU among patients undergoing mitral valvuloplasty cannot be conclusively interpreted due to the small sample size. 

Early postoperative diastolic dysfunction following cardiac surgery poses a significant challenge, with limited understanding of its recognition and management [21,22,23]. Identifying diastolic heart failure in postoperative patients can be particularly challenging given the lack of well-defined diagnostic criteria in the ICU setting where immediate postoperative care is provided. Due to the similarity in clinical presentation, it is crucial to differentiate between diastolic and systolic heart failure, necessitating specialized TEE evaluation, including comprehensive tissue Doppler assessment [21,22,23,24]. Therefore, specialized TEE examination becomes crucial in managing diastolic heart failure postoperatively in the ICU, aiding in formulating tailored treatment plans based on accurate diagnosis [10,14]. The benefit analysis favors specialized TEE pre-CPB for diagnosing ischemic MR severity and assessing the feasibility of conservative mitral surgery. Post CPB, TEE is employed to evaluate residual MR following mitral valvuloplasty or CABG for ischemic MR. Moreover, for patients with concurrent infective endocarditis involving the mitral lesion, specialized TEE proves to be economically efficient. Early postoperative specialized TEE in ICU in patients after MV surgery is usually performed for suspicion of pericardial or pleural collections [24], evaluation of the severity of pulmonary hypertension [25], determining the etiology of an acute hemodynamic dysfunction that suddenly develops immediately postoperatively and is unexplained by other investigation methods, complete evaluation of diastolic performance in order to diagnose postoperative diastolic dysfunction, and when there is no time for performing other invasive investigations [16]. In all of these situations, the benefit ratio was favorable also in our study.

There were some differences noticed regarding the indications of TEE between the results of our study and the literature. 

Published guidelines often overlook certain patient subgroups for whom our study found a favorable benefit ratio for specialized TEE, such as in diagnosing associated valve lesions (aortic or tricuspid) post MV surgical treatment and for triple-valve replacement [1,26]. This discrepancy may stem from the low prevalence of these pathologies in our country due to successful cardiac arthritis prevention programs and accurate preoperative diagnoses facilitated by advanced technologies. Additionally, the utility of immediate postoperative TEE in the ICU for suspected pericardial or pleural collections or assessing residual MR after valvuloplasty is not outlined in guidelines due to low occurrence rates and higher experience levels in mitral valvuloplasty in other countries, where TEE is extensively used intraoperatively. Moreover, routine evaluation of MV prostheses after CPB is presented as a relative indication in the literature but showed limited benefit in our study, possibly due to extensive experience with valve replacement in our country, leading to selective TEE utilization for suspected prosthesis dysfunction [27,28,29].

## 5. Study Limitations

Among the limitations of the study, we can mainly point out the fact that we could not perform TEE in all patients with MV disease undergoing cardiac surgery and the present study did not take into consideration the usefulness of specialized perioperative TEE in patients with minimally invasive and robotic MV surgery or transcatheter aortic valve implantation. Also, intraoperatively, we did not perform 3D TEE evaluation because of the reduced availability of the echocardiograph. On the other hand, the small number of patients in whom MV repair was performed was associated with an NS *p*-value, but the conclusions were similar to those in the reviewed literature.

## 6. Conclusions

Intraoperative TEE is considered the gold standard for some MV surgery interventions, ensuring the safety of surgical procedures, particularly in patients with precarious hemodynamic status. While standardized measurements can generally be performed by an anesthesiologist trained in perioperative TEE, certain situations necessitate a specialized senior cardiologist.

The correlation between surgical diagnoses and those obtained by specialized TEE was notably good and superior to those associated with routine intraoperative TEE, especially in specific scenarios.

Our analysis indicates that the benefits of performing specialized perioperative TEE in MV surgery can be categorized into standard, relative, and uncertain indications, applicable both intraoperatively and immediately postoperatively. Standard intraoperative indications include determining the mechanism and severity of MR, guiding MV valvuloplasty, diagnosing associated valvular lesions post CPB, routine evaluations in triple-valve replacement, and identifying causes of acute, intraoperative, life-threatening hemodynamic dysfunction. Early postoperative specialized TEE in the ICU is strongly indicated for suspected pericardial or pleural effusions, evaluating the severity of residual MR post-mitral valvuloplasty, and establishing the etiology of acute hemodynamic dysfunction.

## Figures and Tables

**Figure 1 diagnostics-14-01095-f001:**
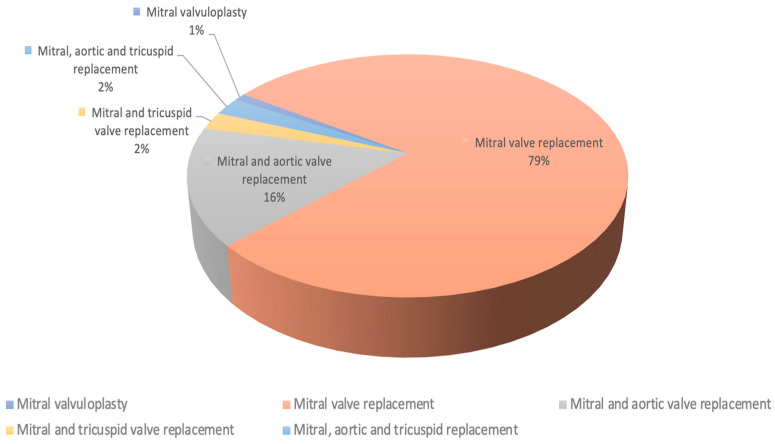
Study group structure depending on the type of the surgery performed.

**Figure 2 diagnostics-14-01095-f002:**
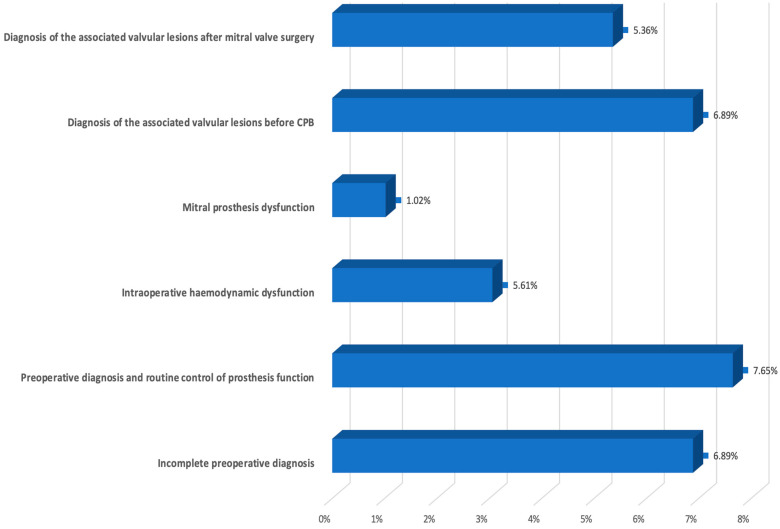
Indications for specialized intraoperative TEE.

**Figure 3 diagnostics-14-01095-f003:**
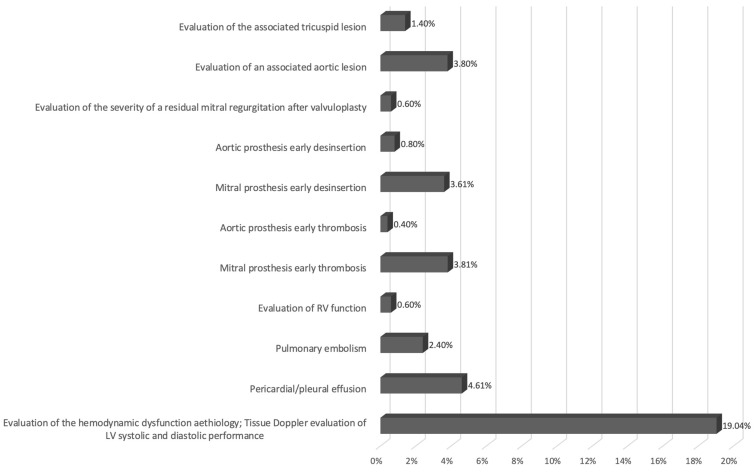
The indications for specialized early postoperative TEE in the ICU.

**Table 1 diagnostics-14-01095-t001:** Intraoperative specialized TEE—indications and implications for therapeutic approach.

Reasons for Performing Intraoperative TEE (%)	Details	Correlation Coefficient between Specialized and Routine TEE Diagnosis	*p*-Value	Benefit Ratio for Specialized TEE	Change in Therapeutic Attitude/Surgical Strategy in Response to the Diagnosis Provided by Specialized TEE
Correcting and completing the preoperative diagnosis of the MV lesion in view of valvuloplasty or MVR (6.89%)	Before CPB	0.72	0.042	lower	No change
Confirmation or denial of associated valvular lesions before MVR (6.89%)	Before CPB	0.23	0.037	higher	Modification of the surgical strategy (associated aortic valve replacement or tricuspid valve surgery)
Evaluating associated valvular lesions that may have been underestimated preoperatively due to the MV lesion (5.36%)	After CPB	0.17	0.021	higher	Modification of the surgical strategy (associated aortic valve replacement or tricuspid valve surgery)
More accurate preoperative diagnosis and routine control of prosthesis function (7.65%)	Before and after CPB	0.75	0.004	lower	No change
Evaluating acute, life-threatening hemodynamic dysfunction after CPB (5.61%)	After CPB	0.26	0.035	higher	Improve filling (filling deficit) Aortic counterpulsation (global LV systolic dysfunction) Medical treatment (RV systolic dysfunction)
Suspicion of prosthesis dysfunction (1.02%)	After CPB	0.67	0.027	lower	No change

**Table 2 diagnostics-14-01095-t002:** Early postoperative specialized TEE—indications, correlation between specialized and routine TEE diagnoses, and implications for therapeutic approach.

Reasons for Performing Early Postoperative TEE in the ICU (41.8%)	Diagnosis Provided by TEE—Details	Correlation Coefficient between Specialized and Routine TEE Diagnosis	*p*-Value	Benefit Ratio for Specialized TEE	Therapeutic Changes Based on the Specialized TEE Diagnosis
Diagnosis of the etiology of acute hemodynamic dysfunction (19.4%)	Filling deficit with severe LV diastolic dysfunction	0.12	0.002	higher	Increase filling fluids
	Severe global LV systolic dysfunction	0.34	0.004	higher	Inotropic support
	New segmental contractility disorders	0.11	0.003	higher	Coronary angiography
	Significant pulmonary hypertension/pulmonary thromboembolism	0.45	0.34	higher	Specific treatment with anticoagulants and vasodilators
	Pericardial or pleural effusion	0.51	0.22	higher	Fluid evacuation
Suspicion of pericardial or pleural effusion (4.61%)	Pericardial or pleural effusion	0.55	0.051	higher	Fluid evacuation
Suspicion of pulmonary embolism (2.40%)	Partial diagnosis	0.12	0.231	lower	CT to confirm diagnosis
Suspicion of early mitral or aortic prosthesis dysfunction (8.62%)	Suspicion of prosthesis early thrombosis	0.48	0.045	lower	Confirmed, no change
	Suspicion of paraprosthetic leakage	0.32	0.005	higher	Reintervention
Evaluation of the severity of a residual MR after mitral valvuloplasty (0.6%)	The same as intraoperatively	0.89	0.007	lower	No change
To assess the severity of the associated valvular lesion known preoperatively (5.21%)	The same as pre- and intraoperatively	0.42	0.003	lower	No change
To assess the RV systolic performance (0.6%)	RV severe dysfunction	0.23	0.235	equal	Medical treatment

**Table 3 diagnostics-14-01095-t003:** Indications of specialized TEE in MV surgery depending on the cost–benefit ratio.

** *Standard indications for specialized TEE* **	Intraoperative indications for specialized TEE	To establish the mechanism and severity of MR
		To guide MV valvuloplasty
		Diagnosis of associated valvular lesions (aortic or tricuspid) by assessing the severity and appropriateness of surgical correction after MV surgical treatment
		Routine evaluation in triple-valve replacement (mitral, aortic, and tricuspid)
		Establishment of causes of acute, intraoperative, life-threatening hemodynamic dysfunction
	Early postoperative indications in the ICU for specialized TEE	Suspicion of pericardial or pleural collections
		Establishing the etiology of acute hemodynamic dysfunction
		To evaluate the severity of a residual mitral insufficiency after mitral valvuloplasty
** *Relative indications for specialized TEE* **	Intraoperative indications for specialized TEE	Preoperative diagnosis in patients with LV systolic dysfunction and valvular lesions associated with the mitral one
	Early postoperative indications in the ICU for specialized TEE	Appreciating the RV systolic performance in triple valve replacement (mitral, aortic, and tricuspid)
** *Uncertain indications for specialized TEE* **	Intraoperative indications for specialized TEE	Preoperative diagnosis of MV lesion or associated valve lesions in patients with poor acoustic window who were not adequately explored preoperatively
		Routine assessment of the prosthesis after CPB
	Early postoperative indications in the ICU for specialized TEE	Suspicion of pulmonary embolism
		Suspicion of thrombosis of mitral or aortic prosthesis
		Diagnosis of the associated valve lesion severity

## Data Availability

All data generated or analyzed during this study are included in this published article.

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
