# Peer review of "Challenges Regarding the Value of Routine Perioperative Transesophageal Echocardiography in Mitral Valve Surgery"

_diagnostics, 2024, doi:10.3390/diagnostics14111095_

Round 1

Reviewer 1 Report

Comments and Suggestions for Authors

This reviewer has the following comments to the Authors:

-       Page 2: Recent guidelines empathized? It should be emphasized.

-       Page 3: Please change “ejection tract of LV” with a more appropriate LVOT (left ventricular outflow tract)

-       Page 4: please specify the meaning of the following acronym: GDPR

-       Page 4, in Ultrasound Methods: there are double brackets open and not closed, please pay attention to this second paragraph of the chapter. Moreover,please rephrase in a more understandable way this line “The surgical interventions were codified by the type of valvular lesion and surgical intervention of the study group structure as is presented in Figure 1”

-       Page 5, statistical analysis: please write better and less schematic this chapter, especially when are specified the type of test for the type of variable. Again there is another double brackets that should be deleted. Moreover, please try to use the same verbal tense in the same chapter (eg. “also we calculate etc…”, where all the rest of the chapter is written in past simple).

-       Please explain better Table 1… it is not very clear what the numbers represent in each line…

-       The work is very prolix, with a terrible English. A list of information, without good connection. Very difficult to read everything.

-       Please ameliorate your English.

Comments on the Quality of English Language

Very bad English, with some grammar errors but overall a difficult form to read.

Reviewer 2 Report

Comments and Suggestions for Authors

The authors have attempted to evaluate their clinical practice with regards to pre- intra- and post-operative TEE to generate recommendations on the cost-effectiveness of using the modality in the clinic.  Unfortunately, I do not believe that this study is novel or additive to the existing literature on the value of TEE.  The manuscript and associated data is confusing and perhaps I am misunderstanding the research in my questions below - if this is the case, then extensive reworking is needed so that the results are more clear.

1. The statistical analysis is unclear - the manuscript focuses on the cost-benefit ratio of TEE with correlations between 'specialized' and 'non-specialized' diagnoses.  However, other statistical tests are specified in the methods and it does not appear that they were used (e.g., chi square tests, t-tests).

2. The calculation of the cost-benefit ratio, done using "software...obtained from the specialized departments of our institute" is unclear.  How is it actually calculated?  How are 'costs' calculated - the dollar amount of the provider's salary/time needed to perform the exam? Is this broken up into each individual component (i.e., the 2 minutes to interrogate residual MR post-repair versus the 20 minutes pre-operative evaluation to assess the lesion)?

3. Table 1: the data in the table are unclear.  What are the data values in the columns?  If they are 'benefit scores', how are they derived?  Where do the numbers in columns 2 and 3 come from?  What are the number ranges in the first column (p-values? and if so what hypothesis do they refer to?)

4. I feel like some of the findings are irrelevant - for example, finding that there is a poor cost-benefit ratio to interrogating a prosthesis for potential dysfunction; if the surgeon has any suspicion whatsoever, it is always beneficial to to the exam regardless of whether the result does not support the suspicion.   What is the true 'cost' associated with this?  The probe is most likely still in/near the patient, the anesthesiologist likely still present, and the amount of time to do the exam very minimal.

4.  The use of the 'Moodle' platform and it's relevance to this study is unclear.  It is described in the methods but not mentioned in the results, yet talked about in the discussion.  How exactly was this used in the study?  How did it actually contribute to the decisions on the use of specialized versus non-specialized TEE?

Comments on the Quality of English Language

Extensive editing for English grammar and syntax is required.

Reviewer 3 Report

Comments and Suggestions for Authors

I would thank the Editor for giving me the opportunity to review the manuscript writen by Iliuta and collegues.

In this paper the authors analized the usefulness of routine specialized perioperative transesophageal echocardiography in patients underzent mitral valve surgery.

First of all I want to congratulate with author for this work, the manuscript is full of informations and very intresting.

However, I have some comment

1)      In my opinion the manuscript is very long and too verbose to explain very simple concepts. The authors spent a lot of sentences, perhaps too much, in arguing the various sections of the manuscript. In some cases, this aspect causes us to lose sight of the true objectives of the manuscript, resulting in superfluous comments. We suggest that authors restructure the document by making it shorter, less verbose and more effective in addressing the main points of the research.

2)      A strong limitation of this article is linked to the fact that a very small number of patients undergoing mitral valve repair is taken into consideration. The 79% of patients underwent mitral valve replacement and 16% combined mitral and aortic valve replacement, threfore the 95% undewent a valve replacement procedure. In this type of patients I suppose that the preoperative indications comes from a specialized cardiologist that previously performed transthoracic echocardiogram + coronary angiogram and/or ct scan. This means that patients should have a precise diagnosis preoperatively. If valve replacement surgery is performed the most important evaluation in OR regards the function of the prosthesis, presence of paravalvular leaks, and ventricular function. All these characteristics are generally well investigated by a non-specialized anesthesiologist. The problem starts to be worrying when mitral valve repair is performed. In this case surgeon should be elucidated about the mechanism of the regurgitation, result of the repair and 3D analysis is often needed. In this regard it appears somewhat incomplete this study because the sample considered is not completely adapted to resolve this debate.

3)      References should be checked.

4)      The discussion section is too long. I strongly recommend reducing the discussion section and improving it by adding a discussion on the use of TEEs in contexts other than those analyzed in the study including minimally invasive surgery,

Ender J, Sgouropoulou S. Value of transesophageal echocardiography (TEE) guidance in minimally invasive mitral valve surgery. Ann Cardiothorac Surg. 2013 Nov;2(6):796-802. doi: 10.3978/j.issn.2225-319X.2013.10.09. PMID: 24349984; PMCID: PMC3857007.

Prempeh ABA, Scherman J, Swanevelder JL. Transesophageal echocardiography in minimally invasive cardiac surgery. Curr Opin Anaesthesiol. 2020 Feb;33(1):83-91. doi:10.1097/ACO.0000000000000807. PMID: 31789893.

 robotic surgery,

Piperata A, Busuttil O, d'Ostrevy N, Jansens JL, Taymoor S, Cuko B, Modine T, Pernot M, Labrousse L. Starting A New Robotic Surgery Program for Mitral Valve Repair. Lessons Learned from The First Nine Months. J Clin Med. 2021 Nov 21;10(22):5439. doi: 10.3390/jcm10225439. PMID: 34830720; PMCID: PMC8674761.

and transcatheter procedures.

Hasnie AA, Parcha V, Hawi R, Trump M, Shetty NS, Ahmed MI, Booker OJ, Arora P, Arora G. Complications Associated With Transesophageal Echocardiography in Transcatheter Structural Cardiac Interventions. J Am Soc Echocardiogr. 2023 Apr;36(4):381-390. doi: 10.1016/j.echo.2022.12.023. Epub 2023 Jan 5. PMID: 36610496; PMCID: PMC10079559.

Please comment and cite these experiences.

I think that once the authors have modified the text with these suggestions the paper will be ready for publication.

Comments on the Quality of English Language

good

Round 2

Reviewer 2 Report

Comments and Suggestions for Authors

The authors addressed all concerns and greatly improved the manuscript.  It is much easier to read and comprehend, and the main objectives are appropriately clear.  The introduction and discussion are well edited, and the methods section has been drastically improved so that it is now easier to understand what has been done.